# Effects of Human Papilloma Virus E6/E7 Oncoproteins on Genomic Structure in Head and Neck Squamous Cell Carcinoma

**DOI:** 10.3390/cancers14246190

**Published:** 2022-12-15

**Authors:** Matthew Uzelac, Armon Barakchi, Varsha Beldona, Daniel John, Jaideep Chakladar, Wei Tse Li, Weg M. Ongkeko

**Affiliations:** 1Division of Otolaryngology-Head and Neck Surgery, Department of Surgery, University of California, San Diego, CA 92093, USA; 2Research Service, VA San Diego Healthcare System, San Diego, CA 92161, USA

**Keywords:** Human Papilloma Virus (HPV), Head and Neck Squamous Cell Carcinoma (HNSCC)

## Abstract

**Simple Summary:**

Human Papilloma Virus (HPV) is known to affect thousands globally. HPV infection can act carcinogenically on host cells with expression of the virus’s E6 and E7 oncoproteins. Within the United States, roughly 70% of oropharyngeal cancers are thought to be HPV induced. Viral genome integration has been well studied, yet genomic effects of the E6 and E7 proteins on other genetic regions remain relatively unidentified. This study characterizes genomic mutation in HPV-infected HNSCC patients with specific regard to host E6 and E7 expression. Individuals with greater presence of these oncoproteins were found to exhibit a greater average of point mutations, particularly on chromosomes 1, 11, and 17. Greater expression of E6 and E7 also correlates to a lesser number of clustered variation events and fewer repeats of copy number segments. Analysis of the genomic effects of HPV may provide additional insight into the pathogenesis of HNSCC.

**Abstract:**

Human Papilloma Virus (HPV) is highly prevalent within the U.S., with studies estimating that over 80% of individuals will contract the virus in their lifetime. HPV is considered a primary risk factor for the development and progression of oropharyngeal cancers. The impact of the HPV virus’s E6 and E7 oncoproteins on cellular signaling pathways and genomic integration has been extensively characterized. Indirect genomic effects; however, remain relatively unidentified. In this study, we analyzed 83 HPV+ Head and Neck Squamous Cell Carcinoma (HNSCC) patients of varying HPV types. Expression counts of the HPV E6 and E7 oncogenes were estimated across samples and correlated with genomic mutational classes. High expression of E6 and E7 oncoproteins was associated with a greater number of total point mutations, especially on chromosomes 1, 11, and 17, which have been implicated in HPV-mediated cancers in previous studies. Samples with high E6 and E7 expression also exhibited more frequent non-clustered structural variation and a lack of clustered variation altogether. Copy number segments were present with fewer number of repeats in high E6 and E7 expression samples, which is known to correlate with decreased expression of affected genes. E6 and E7 expression was associated with increased activity of several cellular pathways associated in oncogenesis and telomere maintenance. In comprehensively characterizing the effects of the HPV oncoproteins on the human genome, potential mechanisms of HNSCC pathogenesis may be further elucidated.

## 1. Introduction

Recent studies estimate that over 80% of individuals in the United States will contract HPV in their lifetime [1]. HPV infection is known to increase an individual’s risk of cancer development, especially cervical squamous cell carcinoma (CESCC) [2,3]. The CDC estimate over 46,000 HPV-associated cancer occurrences within the U.S. from 2014 to 2018 [3]. Approximately 90% of cervical cancers, ~70% of vaginal cancers, and ~60 of penile cancers are attributed to HPV infection. Only recently has oropharyngeal cancer development been associated with HPV infection, with ~70% of cases potentially induced by HPV infection [3].

The American Cancer Society (ACS) approximated that there were ~54,000 new oropharyngeal cancer cases in 2022, with over 11,000 deaths caused by oropharyngeal cancer [4]. Head and Neck Squamous Cell Carcinoma (HNSCC) risk factors are known to include alcohol and tobacco use, as well as HPV infection [5,6,7].

HPV is known to mediate cancer pathogenesis through the E6 and E7 oncogenes. The E6 and E7 proteins target the p53 and pRb proteins, respectively. E6 facilitates the degradation of p53, while E7 inactivates pRb, leading to its eventual degradation [8,9]. Hindered p53 and pRb functionality allows unregulated cellular growth and proliferation, ultimately contributing to carcinogenesis. Expression of the viral E2 gene is known to repress transcription of these E6 and E7 genes. However, upon E1 regulated host-genome integration, the E2 sequence is disrupted, allowing increased E6 and E7 expression [8]. Though effects of the E6 and E7 proteins on the human proteome have previously been characterized, few studies have investigated the specific influence of these proteins on the human genome. Causational mechanisms by which E6 and E7 contribute to cancer development are currently limited to indirect regulation of cell-cycle proteins. With their widely accepted association to carcinogenesis, it may prove useful to characterize other potential cellular effects of the E6 and E7 oncogenes. Direct or indirect genomic alterations have yet to be identified in HPV+ HNSCC cases, though they may act as additional means of cancer promotion.

In understanding the genomic effects of HPV oncoprotein expression on host cells, we could potentially understand the mechanism of why certain patients would be more prone to developing HNSCC after an HPV infection, while other patients do not. Furthermore, understanding this mechanism could also potentially help us identify new treatment targets for personalized treatment of HPV-induced HNSCC.

In this study, 83 HPV+ HNSCC patients from The Cancer Genome Atlas (TCGA) who had been infected with HPV16, HPV18, or HPV33 were analyzed in this study. Patients were categorized based on E6 and E7 gene expression. Mutational profile analyses were conducted to reveal common mutational trends across samples with both high and low expression of these genes. Cellular signaling pathway analyses were performed to identify downstream effects of association with these mutational patterns.

## 2. Materials and Methods

### 2.1. Data Download

STAR 2-Pass transcriptomic RNA-sequencing data, MuTect2 annotated somatic mutation data, STAR-Fusion structural variation data, and ASCAT2 allele-specific copy number segment data were downloaded from the GDC Repository for 83 HNSCC HPV+ patients (https://portal.gdc.cancer.gov/repository) (accessed on 23 July 2022). Corresponding clinical data of these patients were downloaded from the Broad GDAC Firehose (https://gdac.broadinstitute.org/) (accessed on 2 August 2022).

### 2.2. HPV Genome Expression Count Extraction

HPV16, HPV18, and HPV33 reference genome files were downloaded from the NIH’s PaVE (https://pave.niaid.nih.gov/explore/reference_genomes/human_genomes) (accessed on 23 July 2022). Files were processed using the STAR RNA-sequence aligner’s genome generate function [10].

Transcriptomic RNA-sequencing data were annotated with HPV reference genomes, with regard to each patient’s clinically listed HPV type. An out-filter score of 0.5 was used to generate unstranded read counts for the E1–E7 and L1–L2 genes across samples.

### 2.3. E6/E7 Transcripts per Million Calculation

*TPM* was calculated for E6 and E7 genes using the below formula, where *C_g_* denotes the number of reads aligned to gene *g*, *L_g_* denotes the exon lengths of gene *g*, and *N* denotes the total number of protein coding genes.
$$TPM=\frac{(C_{g}*e^{3}/L_{g})*e^{6}}{\sum_{g=1}^{N}\left(C_{g}*e^{3}/L_{g}\right)}$$

### 2.4. Single Base Substitution and Indel Analysis

Annotated somatic mutation data were processed using SigProlierMatrixGenerator [11]. SBS counts were normalized to account for varying chromosome size; counts on a chromosome were divided by the ratio of the ~bp length of that chromosome to the ~bp length of the human genome. Patients were categorized by E6/E7 joint-*TPM*, with *TPM* ≥ 20 considered high expression, and *TPM* < 20 low expression. SBS and indel counts were structured, and corresponding bar charts and heatmaps were created using SigProfilerPlotting.

### 2.5. Structural Variation Analysis

Structural variation data were processed using SigProfilerMatrixGenerator [11]. Patients were categorized by individual E6/E7 *TPM*, with *TPM* ≥ 10 considered high expression, and *TPM* < 10 low expression. Structural variation counts were structured, and corresponding plots were created using SigProfilerPlotting.

### 2.6. Copy Number Variation Analysis

Allele-specific copy number segment data were processed using SigProfilerMatrixGenerator [11]. Patients were categorized by individual E6/E7 *TPM*, with *TPM* ≥ 10 considered high expression, and *TPM* < 10 low expression. Copy number variation counts were structured, and corresponding plots were created using SigProfilerPlotting.

### 2.7. Gene Set Enrichment Analysis

A total of 188 oncogenic and 37 telomeric gene sets were sourced from the Molecular Signaling Database’s (MSigDB) hallmark gene set collection (https://www.gsea-msigdb.org/gsea/msigdb) (accessed on 4 August 2022). Protein coding gene expression counts were analyzed using GSEA software [12,13]. E6/E7 expression served as the phenotype of concern. Patients were categorized by individual E6/E7 *TPM*, with *TPM* ≥ 10 considered high expression, and *TPM* < 10 low expression. Significance of relation was calculated, with *p*-value < 0.05 and q-value < 0.25 used as significance thresholds.

## 3. Results

### 3.1. Chromosomal Point Mutation Frequency

The chromosomes with the greatest frequencies of point mutations were first identified. Single base substitutions (SBS) were extracted across samples. The number and type of substitutions were averaged across patients. SBS counts were normalized by chromosome length to account for varying chromosome size. Chromosome 19 displayed a considerably greater frequency of point mutations compared to others. Chromosomes 1, 11, 17, and 22 also contained relatively high frequencies of mutation (Figure 1).

### 3.2. E6/E7 Expression Association with Genomic Mutation and Structural Variation

E6/E7 high- and low-expression cohorts were analyzed for point mutations by chromosome number. As mentioned above, chromosomes 1, 11, 17, 19, and 22 contained the greatest frequencies of point mutations among samples. Chromosomes 19 and 22 did not exhibit a significant difference in point mutation frequency between high- and low-expression cohorts. Chromosomes 1, 11, and 17 are believed to widely exhibit chromosomal defects in HPV+ cervical squamous cell carcinoma (CESCC) patients [14,15,16]. In samples of high E6/E7 expression, the average numbers of substitutions in chromosomes 1, 11, and 17 were found to be ~347, ~213, and ~190, respectively. In samples of low expression, these values were ~333, ~210, and ~173. Kolmogorov–Smirnov tests revealed these variances to be significant concerning chromosome 1 (*p*-value = 0.020), and chromosome 17 (*p*-value = 3.92 × 10^−4^). Samples of high E6/E7 expression displayed generally greater amounts of point mutations. Notably, C > A and C > T mutations in high-expression samples exceeded those in low-expression samples. T > G mutations were of greater presence in low-expression samples, however (Figure 2).

Indel counts were extracted across samples. The number and type of insertions and deletions were averaged across patients within a cohort. In E6/E7 high-expression samples, deletions > 1 bp in length were of considerably greater presence across chromosomes 1, 11, and 17; high- and low-expression deletion averages were ~18, ~8, ~9 and ~11, ~5, ~6, respectively. Insertion averages were relatively insignificant, as were microhomologous deletions (Figure 3).

Structural variation describes altered regions of DNA much larger in size than point mutations, often introduced by means of recombination^17^. Clustered and non-clustered events were extracted and categorized by event type: deletions, tandem duplications, inversions, and translocations [17,18]. Deletions describe DNA sections that were omitted upon recombination events. Tandem duplications occur when DNA sections are included at one or more repeats. Inversions describe DNA sections of which both strands were recombined in the reverse direction. Translocations involve the migration of DNA sections from one chromosome to another (often homologous). Regarding both the E6 and E7 comparisons, non-clustered average deletions, tandem duplications, and inversions were of far greater presence with high expression. Especially interesting was the near absence of clustered events altogether in high-expression cohorts; low-expression cohorts displayed rather comparable clustered and non-clustered averages, while high-expression cohorts exhibited purely sporadic structural variances. Considering both E6 and E7, high expression correlated with a significant increase in deletions of length 1 Mb–10 Mb, as well as inversions of length > 1 Mb (Figure 4).

Copy number variation is a form of structural variation in which DNA segments are present in varying numbers of repeats across individuals [19]. Repeat numbers within samples were extracted and categorized by event type: heterozygous deletion (HD), heterozygous repeat, and loss of heterozygosity (LOH) [19]. HD describes complete omission of repeats within a chromosome. Heterozygous repeats describe distinct DNA repeat sections of the chromosomes in a homologous pair. LOH occurs when a repeat section of a chromosome is removed and replaced by a corresponding section of the homologous chromosome; LOH is widely believed to correlate with cancer development and progression, as hindered genetic function of a chromosome in a homologous pair can prove considerably harmful [20]. Similar for both E6 and E7 heterozygous comparisons, low expression cohorts contained generally a greater number of repeated segments, with 3–4, 5–8, and 9+ categories being heavily populated. Though many samples of high expression also contain repeats of these lengths, the presence of single repeats (two) were of significantly greater occurrence as compared to low-expression samples. Neither HD nor LOH related significantly to E6 or E7 expression (Figure 5).

### 3.3. E6/E7 Expression Association with Cellular Signaling

The E6 and E7 genes are widely accepted to promote oncogenesis, especially CESCC. The specific effects of these oncogenes on telomere activity and maintenance; however, are understood to a lesser extent [21,22]. Continued maintenance of telomeres may cause unregulated proliferation of tumor cells. It is suspected that E6 expression allows for increased promotion and transcription of the hTERT gene, resulting in greater telomerase activity, and therefore greater genomic stability [22]. To confirm the suspected influence of E6 and E7 expression on carcinogenesis, oncogenic and telomeric pathway activity was analyzed between samples of high and low E6 and E7 expression. Gene read counts were divided into independent E6 and E7 patient cohorts. A total of 188 oncogenic and 37 telomeric gene sets were used a reference of the known genes associated with each signaling pathway. Gene set activity was considered using nominal enrichment scores (NES), calculated based on expression counts of all genes associated within a pathway. A NES describes the extent and direction of which pathways are enriched among cohorts. Between both E6 and E7 cohorts 10 oncogenic gene sets were significantly enriched in samples of high expression (*p*-value < 0.05). Three telomeric gene sets were significantly enriched, with notable false discovery rates (*p*-value < 0.05, q-value < 0.25) (Figure 6).

Enrichment plots were created for each significant gene set, depicting each protein coding gene’s expression influence on enrichment scores. The extent of enrichment is indicated by score magnitude, while direction is indicated by score sign (Figure 7). Notably, genes associated in inhibition of recombinatory DNA pathways were of greater presence in high-expression samples of both E6 and E7.

## 4. Discussion

On average, sample cohorts of higher E6/E7 expression contained a greater number of point mutations, specifically on chromosomes 1, 11, and 17. Mutation within these chromosomes has been affiliated with HPV+ CESCC development and progression, yet remain rather unexplored among HNSCC cases [14,15,16].

Chromosome 17 houses the p53 gene, one of the most important tumor suppressors in all of oncology. Though studies have shown the E6 and E7 genes to affect the p53 protein, the possibility of direct mutation of the p53 gene might also be of significance [23,24]. Other common oncogenes, notably HER2, TOP2A, and TAU, and the double-stranded DNA repair gene, RDM1, are also located on chromosome 17 [25]. Increased mutation of HER2, TOP2A, and TAU might result in highly active proteins, with greater affinity for their cellular targets [26,27,28]. HER2 mutation has been shown to exhibit the potential for resistance to anti-HER2-based therapies [22].

Chromosome 11 has been heavily researched in oncogenic studies, particularly those of breast cancers [29,30]. This chromosome contains a significant tumor-suppressive locus (11p15), often subject to LOH in cancer samples [30]. Mutation of genes within this locus might present similarly to the mutations described above, with lack of tumor-suppressor functionality contributing to continued cell-cycle progression and cellular proliferation.

Chromosome 1 has also seen extensive research among breast cancers [31]. Containing many common sites of translocation and copy number losses, this chromosome is known to contribute rather extensively to carcinogenesis with mutation of specific loci [31,32,33]. Greater structural variation, as seen in cancer samples, would account for increased incidence and progression rates.

Samples of high E6 and E7 expression contained a greater frequency of C > A and C > T mutations. This may be consistent with recent findings regarding APOBEC mediated cytidine deamination [34,35]. APOBEC is believed to be capable of genomic mutagenesis in HNSCC patients, with expression largely induced by HPV infection [34,35]. Greater presence of APOBEC in samples of high E6 and E7 expression may account for the observed increase in cytidine modification. Subsequent analysis investigating APOBEC abundance with relation to E6 and E7 expression may be of significant utility.

Samples of high E6 and E7 expression contained greater amounts of non-clustered structural variation. Significant variation in chromosome structure is recurrent in an array of cancer types, considered as the largest class of all mutation [36,37]. With an increase in deleted segments, tandem duplications, and inversions, samples of high E6 and E7 expression are more prone to oncogenic effect [37,38]. Genes affected by these variations may see decreased promotion or recognition by necessary transcription factors; the possibility of exon damage is also likely [37,38]. In the occurrence of unexpressed or damaged tumor suppressor genes, cell-cycle progression will continue unregulated. In this way, increased expression of the E6 and E7 genes might be of greater contribution to cancer development among high-risk HPV patients.

Worth noting was the near absence of clustered variation events in high-expression cohorts. This might suggest that the E6 and E7 genes especially promote random structural variations, as opposed to confinement in clustered regions of a genome. As described above, mutation in the RDM1 gene or other genes commonly associated in DNA recombination might plausibly account for this randomness [25,39].

Cohorts of high E6 and E7 expression also exhibited fewer copy number repeats, whereas low-expression cohorts contained multiple repeats. Copy number variation has been generally shown to correlate linearly with gene expression [40]. As such, samples that contain few segmented repeats would also display lower expression of the affected genes. Lack of tumor suppressor genes would again encourage cell-cycle unrest and oncogenesis.

Surprisingly, LOH was not of significant alteration between samples of high and low E6 and E7 expression, nor was HD.

The extent of E6 and E7 expression correlated significantly with several oncogenic pathways. This suggests that the degree of E6 and E7 expression will alter the virus’s oncogenic capabilities. Indeed, with greater E6 and E7 abundance, p53 and pRb degradation pathways will see greater activity [8,9]. E6 and E7 expression was also associated with several telomeric pathways. Interestingly, genes associated with inhibition of DNA recombinatory pathways were upregulated in high-expression samples of both E6 and E7. With decreased ability to repair DNA breaks, many of the observed structural variations can plausibly arise. Tangentially, with greater cellular presence, the E6 and E7 genes are more capable of hTERT regulation, ultimately promoting telomere maintenance [21,22]. This process is crucial in sustaining cancer, as telomeres shorten with repeated rounds of genetic duplication and cellular division [41]. With greater telomerase activity, tumor cells are less subject to shortened telomere associated DNA damage.

Ultimately, exploration of specific genomic alteration may be of use in treating HPV+ HNSCC patients. By characterizing the effects of the E6 and E7 oncogenes on the human genome, genetic targets may be identified for treatment. Moreover, investigation of the efficacy of select cancer therapies with regard to HPV infection may be useful. In analyzing the potential correlation of E6 and E7 expression to treatment response, health professionals might be more capable of creating optimal treatment plans for their patients. Effective therapies can be provided, and ineffective therapies can be disregarded. In this way, patients may receive tailored treatment approaches most suitable for their unique conditions.

It should be noted that this study is limited by its inability to discern genomic effects that are distinct for each HPV type. Although HPV16, HPV18, and HPV33 are known to display slight variations in infection and cellular regulation, patients of each type were compiled and analyzed collectively due to insufficient sample sizes. Likewise, neither the extent of viral genomic integration, the anatomical site of tumorigenesis, nor the presence of HNSCC risk factors (alcohol or tobacco use) were corrected for.

## 5. Conclusions

Formulating an understanding of common mutational patterns among HPV patients is beneficial in characterizing the virus’s association to carcinogenesis. Additional research regarding specific genetic targets would be of use, particularly regarding alterations in expression and functionality. Also of use might be investigation of how E6 and E7 genes’ expression correlate with treatment efficacy. With samples of high expression seeing potential correlation to reduced rates of remission for select therapies, optimal treatment plans might be deduced based on the degree of viral effect in a patient.

## Figures and Tables

**Figure 1 cancers-14-06190-f001:**
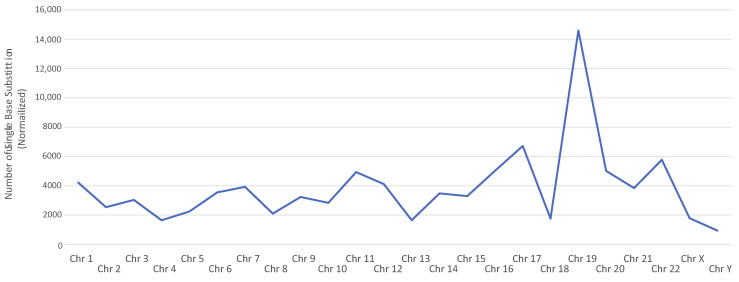
Single base substitution frequency by chromosome. Plot of the normalized average number of point substitutions across samples by chromosome number.

**Figure 2 cancers-14-06190-f002:**
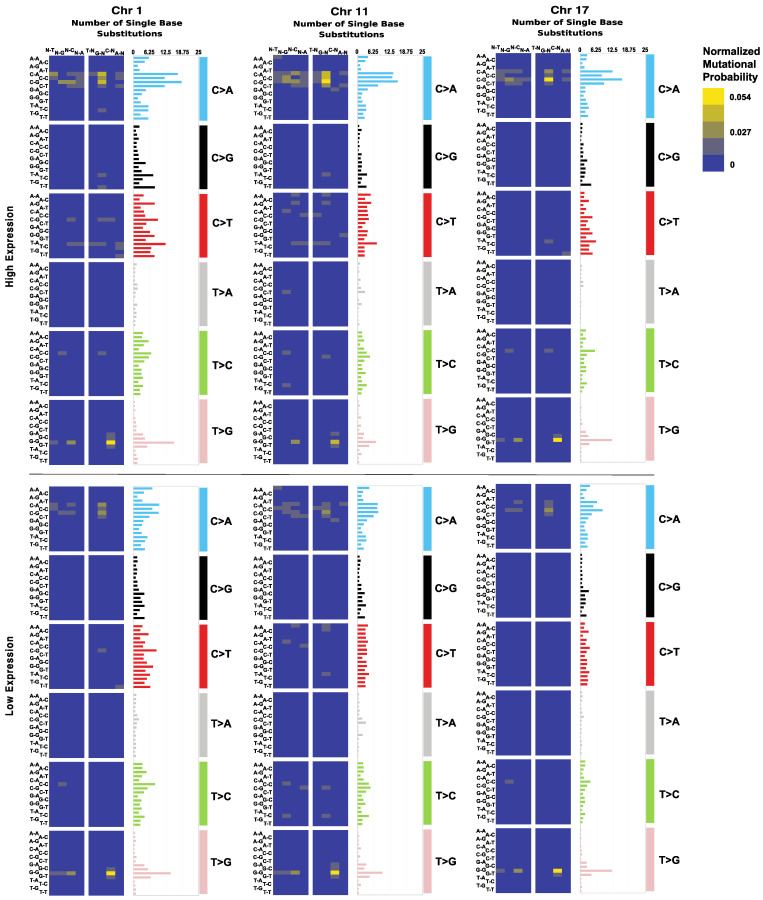
Single base substitution distribution of chromosomes 1, 11, and 17. Bar plots of the average number of point substitutions across samples by E6/E7 high expression and low expression. Heatmaps of the distribution of point mutations by nucleotide type. Patients were categorized by E6/E7 joint-transcripts per million (*TPM*), with *TPM* ≥ 20 considered high expression, and *TPM* < 20 low expression.

**Figure 3 cancers-14-06190-f003:**
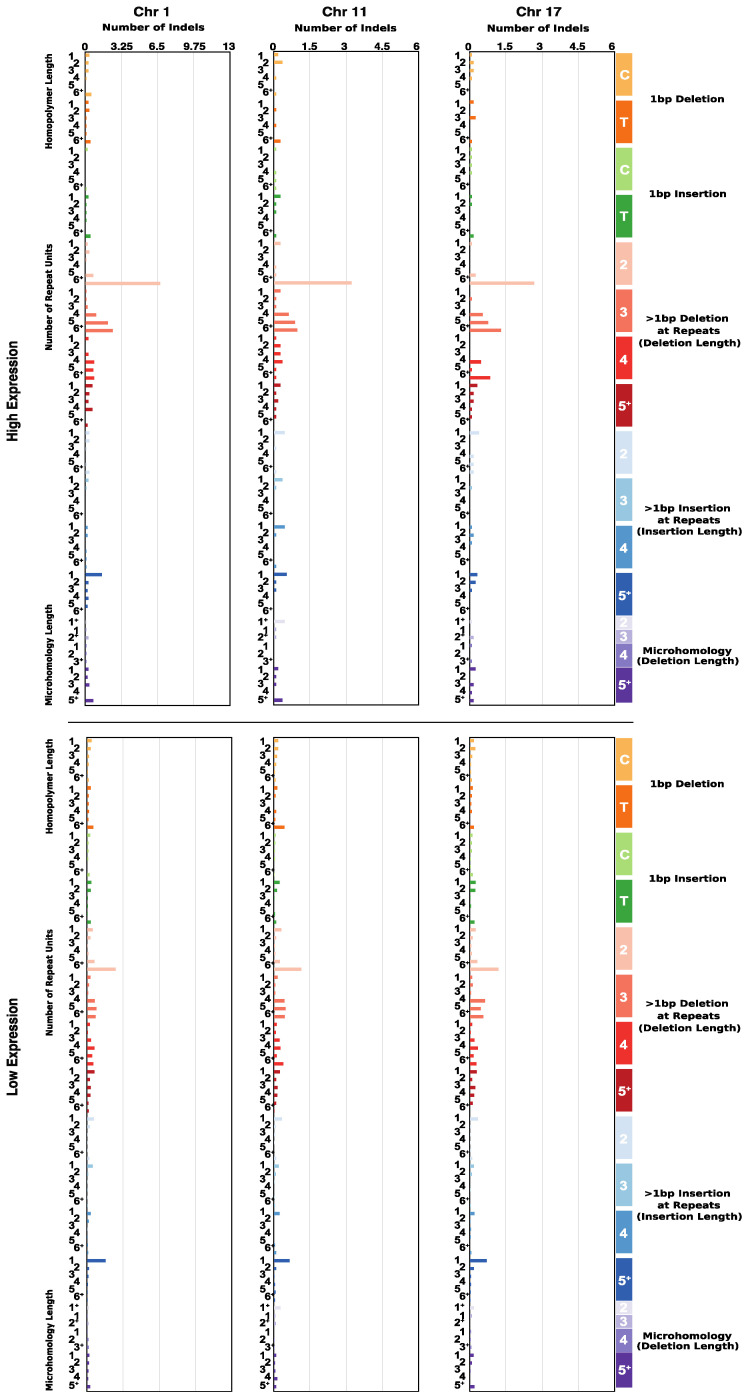
Indel distribution of chromosomes 1, 11, and 17. Bar plots of the average number of point insertions and deletions across samples by E6/E7 high expression and low expression. Patients were categorized by E6/E7 joint-*TPM*, with *TPM* ≥ 20 considered high expression, and *TPM* < 20 low expression.

**Figure 4 cancers-14-06190-f004:**
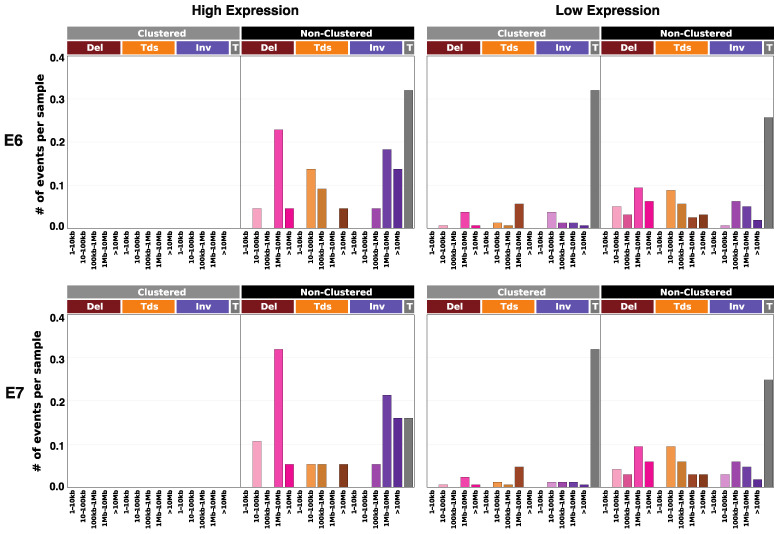
E6/E7 expression correlation to structural variation. Bar plots of the average number of variances across samples by E6/E7 high expression and low expression. Both clustered and non-clustered events are shown, classified as deletions (Del), tandem duplications (Tds), inversions (Inv), and translocations (T). Patients were categorized by individual E6/E7 *TPM*, with *TPM* ≥ 10 considered high expression, and *TPM* < 10 low expression.

**Figure 5 cancers-14-06190-f005:**
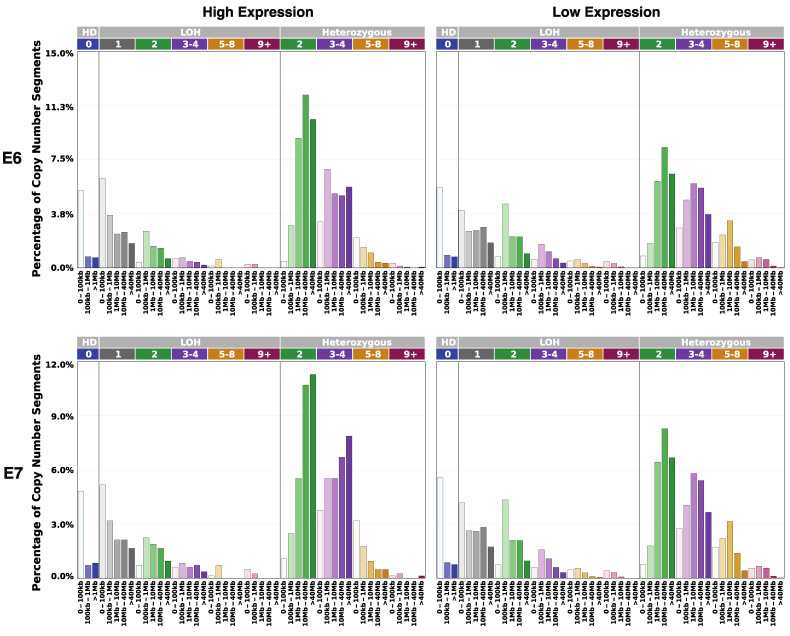
E6/E7 expression correlation with copy number variation. Bar plots of the percentage of segments by length across samples by E6/E7 high expression and low expression. Repeat numbers are shown, classified as heterozygous deletion (HD), loss of heterozygosity (LOH), and heterozygous repeat). Patients were categorized by individual E6/E7 *TPM*, with *TPM* ≥ 10 considered high expression, and *TPM* < 10 low expression.

**Figure 6 cancers-14-06190-f006:**
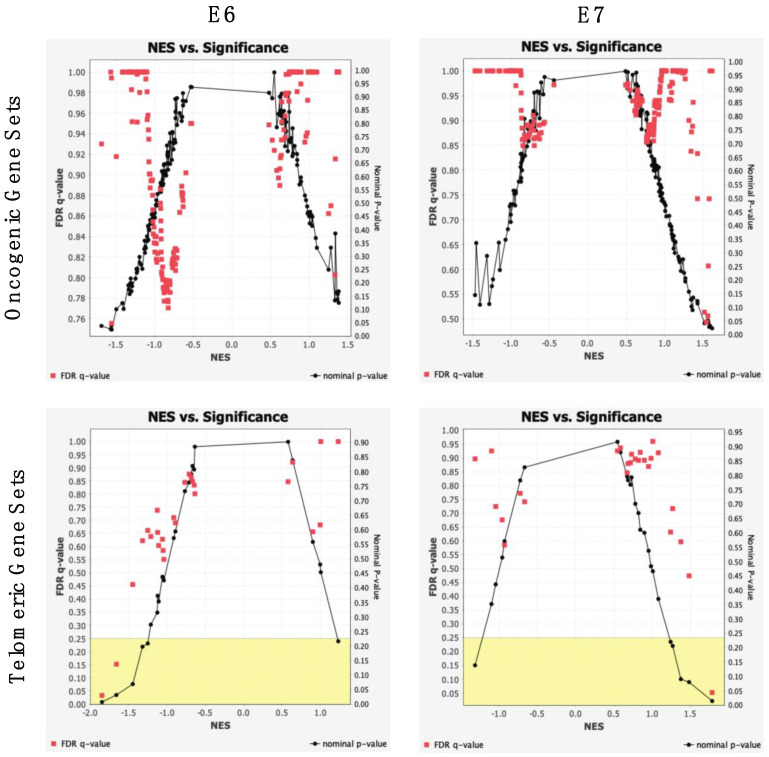
Oncogenic and telomeric signaling pathway significance. Significance curves of oncogenic and telomeric gene sets enriched in samples of E6/E7 high expression and low expression. Nominal enrichment scores (NESs) describe the extent of which gene sets are dysregulated across cohorts. Corresponding nominal p-values and FDR q-values are shown, with *p*-value < 0.05 and q-value < 0.25 (yellow) used as significance thresholds. Patients were categorized by individual E6/E7 *TPM*, with *TPM* ≥ 10 considered high expression, and *TPM* < 10 low expression.

**Figure 7 cancers-14-06190-f007:**
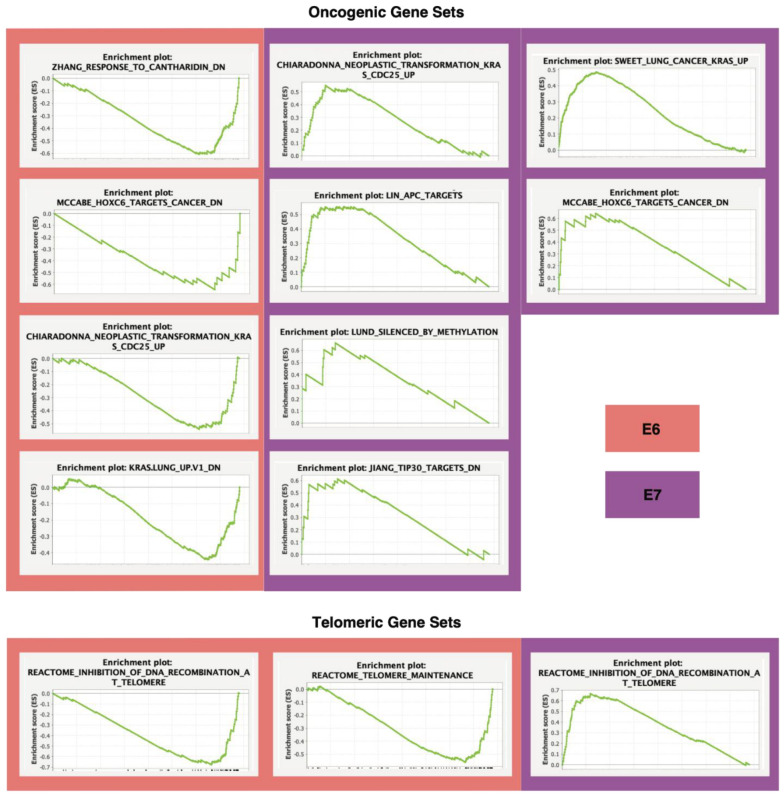
Significant oncogenic and telomeric signaling pathways. Enrichment plots of several significant oncogenic and telomeric gene sets by E6/E7 expression. Enrichment scores (ESs) describe the extent of which gene sets are dysregulated across cohorts. Patients were categorized by individual E6 (pink) and E7 (purple) *TPM*, with *TPM* ≥ 10 considered high expression, and *TPM* < 10 low expression.

## Data Availability

All TCGA data can be accessed online through the TCGA data portal.

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
