# Peer review of "Effects of Human Papilloma Virus E6/E7 Oncoproteins on Genomic Structure in Head and Neck Squamous Cell Carcinoma"

_cancers, 2022, doi:10.3390/cancers14246190_

Round 1

Reviewer 1 Report

Uzelac et al. utilize the TGCA to investigate the effects of HPV infection on the host genomes in HNSCC samples. They correlate viral oncoprotein expression with alterations in DNA, finding that high E6 and E7 levels correlate with deletions and inversions, whereas they had no correlation with loss of heterozygosity or heterozygous deletions. 

In cervical cancers, it is accepted that >98% of HPV positive cancers are integrated and driven by E6 and E7 expression. However in HNSCC, the genome status of HPV is less clear, and the proportion of integrated cancers reported is varied - from 40 to 100%, depending on technique applied. In this study, has the TCGA data been stratified according to episomal/integrated? The tumor types also respond differently to treatment, so it would be interesting to investigate whether their findings relate to patient outcomes.

Overall, the study is straightforward and interesting. I would appreciate more discussion with respect to the relevance to patient treatments and “personalized medicine”.

Reviewer 2 Report

The authors conducted a study to describe the effect of the HPV-oncoproteins E6 and E7 on human genome in head and neck cancers by analysing somatic mutation, structural variation, and copy number segment modifications in high and low expression samples of E6 and E7 genes. 

The manuscript is well written, with nice introduction, clear materials and methods part, concise and well-organized results and finally a large discussion linking their results to the current knowledge in this field.

In the results paragraph, the authors reported that single base substitution frequency was the most prevalent in chromosome 19. However by comparing with E6/E7 expression level, they skipped this chromosome. Is it because no difference were found ? please precised in the manuscript. In this same paragraph, the authors found that C>T and C>A mutations were higher in high expression samples. The authors should discuss about the possibility of the APOBEC effect on these mutations (mainly C>T) in the discussion paragraph, as it was already reported in HNSCC HPV+. 

The authors  included HPV + HNSCC with HPV16, HPV18 and HPV33. However, in the results part they combined all the different HPV types together. Could you explain why and add a limitation in the discussion paper. Indeed, we know from the literratture that the affinity of E6 and E7 for p53 and pRb, respectively, can change between hrHPV. It may also impact their effect on host genome modifications. 

In HPV-induced cancers (cervical, anal and also HNSCC) and precancerous lesions , gain of 3q and gain of 1pq are frequently reported. Does the author have any explanation why they did not find any modifications in that chromosome? 

In all the figure, it would help to add a scale color. 
